# MixKD: Towards Efficient Distillation of Large-scale Language Models

**Kevin J Liang**[1,2][*]**, Weituo Hao**[1][*]**, Dinghan Shen**[3]**, Yufan Zhou**[4]**, Weizhu Chen**[3]**,
Changyou Chen**[4]**, Lawrence Carin**[1]
[1]Duke University    [2]Facebook AI    [3]Microsoft Dynamics 365 AI
[4]State University of New York at Buffalo
{kevin.liang, weituo.hao}@duke.edu

## Abstract

Large-scale language models have recently demonstrated impressive empirical performance. Nevertheless, the improved results are attained at the price of bigger models, more power consumption, and slower inference, which hinder their applicability to low-resource (both memory and computation) platforms. Knowledge distillation (KD) has been demonstrated as an effective framework for compressing such big models. However, large-scale neural network systems are prone to memorize training instances, and thus tend to make inconsistent predictions when the data distribution is altered slightly. Moreover, the student model has few opportunities to request useful information from the teacher model when there is limited task-specific data available. To address these issues, we propose *MixKD*, a data-agnostic distillation framework that leverages mixup, a simple yet efficient data augmentation approach, to endow the resulting model with stronger generalization ability. Concretely, in addition to the original training examples, the student model is encouraged to mimic the teacher's behavior on the linear interpolation of example pairs as well. We prove from a theoretical perspective that under reasonable conditions *MixKD* gives rise to a smaller gap between the generalization error and the empirical error. To verify its effectiveness, we conduct experiments on the GLUE benchmark, where *MixKD* consistently leads to significant gains over the standard KD training, and outperforms several competitive baselines. Experiments under a limited-data setting and ablation studies further demonstrate the advantages of the proposed approach.

## 1 Introduction

Recent language models (LM) pre-trained on large-scale unlabeled text corpora in a self-supervised manner have significantly advanced the state of the art across a wide variety of natural language processing (NLP) tasks (Devlin et al., 2018; Liu et al., 2019c; Yang et al., 2019; Joshi et al., 2020; Sun et al., 2019b; Clark et al., 2020; Lewis et al., 2019; Bao et al., 2020). After the LM pre-training stage, the resulting parameters can be fine-tuned to different downstream tasks. While these models have yielded impressive results, they typically have millions, if not billions, of parameters, and thus can be very expensive from storage and computational standpoints. Additionally, during deployment, such large models can require a lot of time to process even a single sample. In settings where computation may be limited (*e.g.* mobile, edge devices), such characteristics may preclude such powerful models from deployment entirely.

One promising strategy to compress and accelerate large-scale language models is knowledge distillation (Zhao et al., 2019; Tang et al., 2019; Sun et al., 2020). The key idea is to train a smaller model (a "student") to mimic the behavior of the larger, stronger-performing, but perhaps less practical model (the "teacher"), thus achieving similar performance with a faster, lighter-weight model. A simple but powerful method of achieving this is to use the output probability logits produced by the teacher model as soft labels for training the student (Hinton et al., 2015). With higher entropy than one-hot labels, these soft labels contain more information for the student model to learn from.

---

[*]Equal contribution

Previous efforts on distilling large-scale LMs mainly focus on designing better training objectives, such as matching intermediate representations (Sun et al., 2019a; Mukherjee & Awadallah, 2019), learning multiple tasks together (Liu et al., 2019a), or leveraging the distillation objective during the pre-training stage (Jiao et al., 2019; Sanh et al., 2019). However, much less effort has been made to enrich task-specific data, a potentially vital component of the knowledge distillation procedure. In particular, tasks with fewer data samples provide less opportunity for the student model to learn from the teacher. Even with a well-designed training objective, the student model is still prone to overfitting, despite effectively mimicking the teacher network on the available data.

In response to these limitations, we propose improving the value of knowledge distillation by using data augmentation to generate additional samples from the available task-specific data. These augmented samples are further processed by the teacher network to produce additional soft labels, providing the student model more data to learn from a large-scale LM. Intuitively, this is akin to a student learning more from a teacher by asking more questions to further probe the teacher's answers and thoughts. In particular, we demonstrate that mixup (Zhang et al., 2018) can significantly improve knowledge distillation's effectiveness, and we show with a theoretical framework why this is the case. We call our framework *MixKD*.

We conduct experiments on 6 GLUE datasets (Wang et al., 2019) across a variety of task types, demonstrating that *MixKD* significantly outperforms knowledge distillation (Hinton et al., 2015) and other previous methods that compress large-scale language models. In particular, we show that our method is especially effective when the number of available task data samples is small, substantially improving the potency of knowledge distillation. We also visualize representations learned with and without *MixKD* to show the value of interpolated distillation samples, perform a series of ablation and hyperparameter sensitivity studies, and demonstrate the superiority of *MixKD* over other BERT data augmentation strategies.

## 2 RELATED WORK

### 2.1 MODEL COMPRESSION

Compressing large-scale language models, such as BERT, has attracted significant attention recently. Knowledge distillation has been demonstrated as an effective approach, which can be leveraged during both the pre-training and task-specific fine-tuning stages. Prior research efforts mainly focus on improving the training objectives to benefit the distillation process. Specifically, Turc et al. (2019) advocate that task-specific knowledge distillation can be improved by first pre-training the student model. It is shown by Clark et al. (2019) that a multi-task BERT model can be learned by distilling from multiple single-task teachers. Liu et al. (2019b) propose learning a stronger student model by distilling knowledge from an ensemble of BERT models. Patient knowledge distillation (PKD), introduced by Sun et al. (2019a), encourages the student model to mimic the teacher's intermediate layers in addition to output logits. DistilBERT (Sanh et al., 2019) reduces the depth of BERT model by a factor of 2 via knowledge distillation during the pre-training stage. In this work, we evaluate *MixKD* on the case of task-specific knowledge distillation. Notably, it can be extended to the pre-training stage as well, which we leave for future work. Moreover, our method can be flexibly integrated with different KD training objectives (described above) to obtain even better results. However, we utilize the BERT-base model as the testbed in this paper without loss of generality.

### 2.2 DATA AUGMENTATION IN NLP

Data augmentation (DA) has been studied extensively in computer vision as a powerful technique to incorporate prior knowledge of invariances and improve the robustness of learned models (Simard et al., 1998; 2003; Krizhevsky et al., 2012). Recently, it has also been applied and shown effective on natural language data. Many approaches can be categorized as label-preserving transformations, which essentially produce neighbors around a training example that maintain its original label. For example, EDA (Wei & Zou, 2019) propose using various rule-based operations such as synonym replacement, word insertion, swap or deletion to obtain augmented samples. Back-translation (Yu et al., 2018; Xie et al., 2019) is another popular approach belonging to this type, which relies on pre-trained translation models. Additionally, methods based on paraphrase generation have also been leveraged from the data augmentation perspective (Kumar et al., 2019). On the other hand, label-altering techniques like mixup (Zhang et al., 2018) have also been proposed for language (Guo et al., 2019; Chen et al., 2020), producing interpolated inputs and labels for the models predict. The

proposed *MixKD* framework leverages the ability of mixup to facilitate the student learning more information from the teacher. It is worth noting that *MixKD* can be combined with arbitrary label-preserving DA modules. Back-translation is employed as a special case here, and we believe other advanced label-preserving transformations developed in the future can benefit the *MixKD* approach as well.

## 2.3 MIXUP

Mixup (Zhang et al., 2018) is a popular data augmentation strategy to increase model generalizability and robustness by training on convex combinations of pairs of inputs and labels $(x_i, y_i)$ and $(x_j, y_j)$:

$$x' = \lambda x_i + (1 - \lambda)x_j \tag{1}$$

$$y' = \lambda y_i + (1 - \lambda)y_j \tag{2}$$

with $\lambda \in [0, 1]$ and $(x', y')$ being the resulting virtual training example. This concept of interpolating samples was later generalized with Manifold mixup (Verma et al., 2019a) and also found to be effective in semi-supervised learning settings (Verma et al., 2019b;c; Berthelot et al., 2019b;a). Other strategies include mixing together samples resulting from chaining together other augmentation techniques (Hendrycks et al., 2020), or replacing linear interpolation with the cutting and pasting of patches (Yun et al., 2019).

## 3 METHODOLOGY

### 3.1 PRELIMINARIES

In NLP, an input sample $i$ is often represented as a vector of tokens $\mathbf{w}_i = \{w_{i,1}, w_{i,2}, ..., w_{i,T}\}$, with each token $w_{i,t} \in \mathbb{R}^V$ a one-hot vector often representing words (but also possibly subwords, punctuation, or special tokens) and $V$ being the vocabulary size. These discrete tokens are then mapped to word embeddings $\mathbf{x}_i = \{x_{i,1}, x_{i,2}, ..., x_{i,T}\}$, which serve as input to the machine learning model $f$. For supervised classification problems, a one-hot label $y_i \in \mathbb{R}^C$ indicates the ground-truth class of $\mathbf{x}_i$ out of $C$ possible classes. The parameters $\theta$ of $f$ are optimized with some form of stochastic gradient descent so that the output of the model $f(\mathbf{x}_i) \in \mathbb{R}^C$ is as close to $y_i$ as possible, with cross-entropy as the most common loss function:

$$\mathcal{L}_{\text{MLE}} = -\frac{1}{n} \sum_i^n y_i \cdot \log(f(\mathbf{x}_i)) \tag{3}$$

where $n$ is the number of samples, and $\cdot$ is the dot product.

### 3.2 KNOWLEDGE DISTILLATION FOR BERT

Consider two models $f$ and $g$ parameterized by $\theta_T$ and $\theta_S$, respectively, with $|\theta_T| \gg |\theta_S|$. Given enough training data and sufficient optimization, $f$ is likely to yield better accuracy than $g$, due to higher modeling capacity, but may be too bulky or slow for certain applications. Being smaller in size, $g$ is more likely to satisfy operational constraints, but its weaker performance can be seen as a disadvantage. To improve $g$, we can use the output prediction $f(\mathbf{x}_i)$ on input $\mathbf{x}_i$ as extra supervision for $g$ to learn from, seeking to match $g(\mathbf{x}_i)$ with $f(\mathbf{x}_i)$. Given these roles, we refer to $g$ as the student model and $f$ as the teacher model.

While there are a number of recent large-scale language models driving the state of the art, we focus here on BERT (Devlin et al., 2018) models. Following Sun et al. (2019a), we use the notation $\text{BERT}_k$ to indicate a BERT model with $k$ Transformer (Vaswani et al., 2017) layers. While powerful, BERT models also tend to be quite large; for example, the default `bert-base-uncased` ($\text{BERT}_{12}$) has ~110M parameters. Reducing the number of layers (*e.g.* using $\text{BERT}_3$) makes such models significantly more portable and efficient, but at the expense of accuracy. With a knowledge distillation set-up, however, we aim to reduce this loss in performance.

### 3.3 MIXUP DATA AUGMENTATION FOR KNOWLEDGE DISTILLATION

While knowledge distillation can be a powerful technique, if the size of the available data is small, then the student has only limited opportunities to learn from the teacher. This may make it much harder for knowledge distillation to close the gap between student and teacher model performance. To correct this, we propose using data augmentation for knowledge distillation. While data augmentation (Yu et al., 2018; Xie et al., 2019; Yun et al., 2019; Kumar et al., 2019; Hendrycks et al.,

2020; Shen et al., 2020; Qu et al., 2020) is a commonly used technique across machine learning for increasing training samples, robustness, and overall performance, a limited modeling capacity constrains the representations the student is capable of learning on its own. Instead, we propose using the augmented samples to further query the teacher model, whose large size often allows it to learn more powerful features.

While many different data augmentation strategies have been proposed for NLP, we focus on mixup (Zhang et al., 2018) for generating additional samples to learn from the teacher. Mixup's vicinal risk minimization tends to result in smoother decision boundaries and better generalization, while also being cheaper to compute than methods such as backtranslation (Yu et al., 2018; Xie et al., 2019). Mixup was initially proposed for continuous data, where interpolations between data points remain in-domain; its efficacy was demonstrated primarily on image data, but examples in speech recognition and tabular data were also shown to demonstrate generality.

Directly applying mixup to NLP is not quite as straightforward as it is for images, as language commonly consists of sentences of variable length, each comprised of discrete word tokens. Since performing mixup directly on the word tokens doesn't result in valid language inputs, we instead perform mixup on the word embeddings at each time step $x_{i,t}$ (Guo et al., 2019). This can be interpreted as a special case of Manifold mixup Verma et al. (2019a), where the mixing layer is set to the embedding layer. In other words, mixup samples are generated as:

$$x'_{i,t} = \lambda x_{i,t} + (1 - \lambda)x_{j,t} \quad \forall t \tag{4}$$

$$y'_i = \lambda y_i + (1 - \lambda)y_j \tag{5}$$

with $\lambda \in [0, 1]$; random sampling of $\lambda$ from a Uniform or Beta distribution are common choices. Note that we index the augmented sample with $i$ regardless of the value of $\lambda$. Sentence length variability can be mitigated by grouping mixup pairs by length. Alternatively, padding is a common technique for setting a consistent input length across samples; thus, if $\mathbf{x}^{(i)}$ contains more word tokens than $\mathbf{x}^{(j)}$, then the extra word embeddings are mixed up with zero paddings. We find this approach to be effective, while also being much simpler to implement.

We query the teacher model with the generated mixup sample $\mathbf{x}'_i$, producing output prediction $f(\mathbf{x}'_i)$. The student is encouraged to imitate this prediction on the same input, by minimizing the objective:

$$\mathcal{L}_{\text{TMKD}} = d(f(\mathbf{x}'_i), g(\mathbf{x}'_i)) \tag{6}$$

where $d(\cdot, \cdot)$ is a distance metric for distillation, with temperature-adjusted cross-entropy and mean square error (MSE) being common choices.

Since we have the mixup samples already generated (with an easy-to-generate interpolated pseudolabel $y'_i$), we can also train the student model on these augmented data samples in the usual way, with a cross-entropy objective:

$$\mathcal{L}_{\text{SM}} = -\frac{1}{n} \sum_i^n y'_i \cdot \log(g(\mathbf{x}'_i)) \tag{7}$$

Our final objective for *MixKD* is a sum of the original data cross-entropy loss, student cross-entropy loss on the mixup samples, and knowledge distillation from the teacher on the mixup samples:

$$\mathcal{L} = \mathcal{L}_{\text{MLE}} + \alpha_{\text{SM}}\mathcal{L}_{\text{SM}} + \alpha_{\text{TMKD}}\mathcal{L}_{\text{TMKD}} \tag{8}$$

where $\alpha_{\text{SM}}$ and $\alpha_{\text{TMKD}}$ are hyperparameters weighting the loss terms.

### 3.4 THEORETICAL ANALYSIS

We develop a theoretical foundation for the proposed framework. We wish to prove that by adopting data augmentation for knowledge distillation, one can achieve $i$) a smaller gap between generalization error and empirical error, and $ii$) better generalization.

To this end, assume the original training data $\{\mathbf{x}_i\}_{i=1}^n$ are sampled i.i.d. from the true data distribution $p(\mathbf{x})$, and the augmented data distribution by mixup is denoted as $q(\mathbf{x})$ (apparently $p$ and $q$ are dependent). Let $f$ be the teacher function, and $g \in \mathcal{G}$ be the learnable student function. Denote the loss function to learn $g$ as $l(\cdot, \cdot)$[1]. The population risk w.r.t. $p(\mathbf{x})$ is defined as $\mathcal{R}(f, g, p) =$

---

[1]This is essentially the same as $\mathcal{L}$ in equation 8. We use a different notation $l(f(\mathbf{x}), g(\mathbf{x}))$ to explicitly spell out the two data-wise arguments $f(\mathbf{x})$ and $g(\mathbf{x})$.

$\mathbb{E}_{\mathbf{x} \sim p(\mathbf{x})} [l(f(\mathbf{x}), g(\mathbf{x}))]$, and the empirical risk as $\mathcal{R}_{emp}(f, g, \{\mathbf{x}_i\}_{i=1}^n) = \frac{1}{n} \sum_{i=1}^n l(f(\mathbf{x}_i), g(\mathbf{x}_i))$. A classic statement for generalization is the following: with at least $1 - \delta$ probability, we have

$$\mathcal{R}(f, g_p, p) - \mathcal{R}_{emp}(f, g_p, \{\mathbf{x}_i\}_{i=1}^n) \leq \epsilon, \tag{9}$$

where $\epsilon > 0$, and we have used $g_p$ to indicate that the function is learned based on $p(\mathbf{x})$. Note different training data would correspond to a different error $\epsilon$ in equation 9. We use $\epsilon_p$ to denote the minimum value over all $\epsilon$'s satisfying equation 9. Similarly, we can replace $p$ with $q$, and $\{\mathbf{x}_i\}_{i=1}^n$ with $\{\mathbf{x}_i\}_{i=1}^a \cup \{\mathbf{x}_i'\}_{i=1}^b$ in equation 9 in the data-augmentation case. In this case, the student function is learned based on both the training data and augmented data, which we denote as $g^*$. Similarly, we also have a corresponding minimum error, which we denote as $\epsilon^*$. Consequently, our goal of better generalization corresponds to proving $\mathcal{R}(f, g^*, p) \leq \mathcal{R}(f, g_p, p)$, and the goal of a smaller gap corresponds to proving $\epsilon^* \leq \epsilon_p$. In our theoretical results, we will give conditions when these goals are achievable. First, we consider the following three cases about the joint data $\mathbf{X} \triangleq \{\mathbf{x}_i\}_{i=1}^a \cup \{\mathbf{x}_i'\}_{i=1}^b$ and the function class $\mathcal{G}$:

- Case 1: There exists a distribution $\tilde{p}$ such that $\mathbf{X}$ are i.i.d. samples from it[2]; $\mathcal{G}$ is a finite set.
- Case 2: There exists $\tilde{p}$ such that $\mathbf{X}$ are i.i.d. samples from it; $\mathcal{G}$ is an infinite set.
- Case 3: There does not exist a distribution $\tilde{p}$ such that $\mathbf{X}$ are i.i.d. samples from it.

Our theoretical results are summarized in Theorems 1-3, which state that with enough augmented data, our method can achieve smaller generalization errors. Proofs are given in the Appendix.

**Theorem 1** *Assume the loss function $l(\cdot, \cdot)$ is upper bounded by $M > 0$. Under Case 1, there exists a constant $c > 0$ such that if*

$$b \geq \frac{M^2 \log(|\mathcal{G}|/\delta)}{c} - a$$

*then*

$$\epsilon^* \leq \epsilon_p$$

*where $\epsilon^*$ and $\epsilon_p$ denote the minimal generalization gaps one can achieve with or without augmented data, with at least $1 - \delta$ probability. If further assuming a better empirical risk with data augmentation (which is usually the case in practice), i.e., $\mathcal{R}_{emp}(f, g^*, \{\mathbf{x}_i\}_{i=1}^a \cup \{\mathbf{x}_i'\}_{i=1}^b) \leq \mathcal{R}_{emp}(f, g_p, \{\mathbf{x}_i\}_{i=1}^n)$, we have*

$$\mathcal{R}(f, g^*, p) \leq \mathcal{R}(f, g_p, p)$$

**Theorem 2** *Assume the loss function $l(\cdot, \cdot)$ is upper bounded by $M > 0$ and Lipschitz continuous. Fix the probability parameter $\delta$. Under Case 2, there exists a constant $c > 0$ such that if*

$$b \geq \frac{M^2 \log(1/\delta)}{c} - a$$

*then*

$$\epsilon^* \leq \epsilon_p$$

*where $\epsilon^*$ and $\epsilon_p$ denote the minimal generalization gaps one can achieve with or without augmented data, with at least $1 - \delta$ probability. If further assuming a better empirical risk with data augmentation, i.e., $\mathcal{R}_{emp}(f, g^*, \{\mathbf{x}_i\}_{i=1}^a \cup \{\mathbf{x}_i'\}_{i=1}^b) \leq \mathcal{R}_{emp}(f, g_p, \{\mathbf{x}_i\}_{i=1}^n)$, we have*

$$\mathcal{R}(f, g^*, p) \leq \mathcal{R}(f, g_p, p)$$

A more interesting setting is Case 3. Our result is based on Baxter (2000), which studies learning from different and possibly correlated distributions.

**Theorem 3** *Assume the loss function $l(\cdot, \cdot)$ is upper bounded. Under Case 3, there exists constants $c_1, c_2, c_3 > 0$ such that if*

$$b \geq \frac{a \log(4/\delta)}{c_1 a - c_2} \text{ and } a \geq c_3$$

---

[2]We make such an assumption because $\mathbf{x}_i$ and $\mathbf{x}_i'$ are dependent, thus existence of $\tilde{p}$ is unknown.

| Model | SST-2 | MRPC | QQP | MNLI-m | QNLI | RTE |
|---|---|---|---|---|---|---|
| $\mathrm{BERT}_{12}$ | 92.20 | 90.53/86.52 | 88.21/91.25 | 84.12 | 91.32 | 77.98 |
| $\mathrm{DistilBERT}_6$ | 91.3 | 87.5/82.4 | —-/88.5 | 82.2 | **89.2** | 59.9 |
| $\mathrm{BERT}_6$-FT | 90.94 | 88.54/83.82 | 87.16/90.43 | 81.28 | 88.25 | 66.43 |
| $\mathrm{BERT}_6$-TMKD | 91.63 | 88.93/83.82 | 86.60/90.27 | 81.49 | 88.71 | 65.34 |
| $\mathrm{BERT}_6$-SM+TMKD | 91.17 | 89.30/84.31 | 87.19/90.56 | 82.02 | 88.63 | 65.34 |
| $\mathrm{BERT}_6$-FT+BT | 91.74 | **89.60/84.80** | 87.06/90.39 | 82.10 | 87.68 | 67.51 |
| $\mathrm{BERT}_6$-TMKD+BT | 91.86 | 89.52/84.56 | 87.15/90.59 | 82.17 | 88.38 | **69.98** |
| $\mathrm{BERT}_6$-SM+TMKD+BT | **92.09** | 89.22/84.07 | **87.57/90.78** | **82.53** | **88.82** | 67.87 |
| $\mathrm{BERT}_3$-FT | 87.16 | 81.68/71.08 | 84.99/88.65 | 75.55 | 83.98 | 58.48 |
| $\mathrm{BERT}_3$-TMKD | 88.76 | 81.62/71.08 | 83.27/87.80 | 75.73 | 84.26 | 58.48 |
| $\mathrm{BERT}_3$-SM+TMKD | 88.99 | 81.73/71.08 | 84.47/88.37 | 75.52 | 84.24 | 59.57 |
| $\mathrm{BERT}_3$-FT+BT | 88.88 | 83.36/74.26 | 85.31/88.81 | 76.88 | 83.67 | 59.21 |
| $\mathrm{BERT}_3$-TMKD+BT | 89.79 | **84.46/75.74** | 85.17/89.00 | 77.19 | 84.68 | **62.82** |
| $\mathrm{BERT}_3$-SM+TMKD+BT | **90.37** | 84.14/75.25 | **85.56/89.09** | **77.52** | **84.83** | 60.65 |

Table 1: GLUE dev set results. We report the results of our $\mathrm{BERT}_{12}$ teacher model, the 6-layer DistilBERT, and 3- and 6-layer *MixKD* student models with various ablations. DistilBERT results taken from Sanh et al. (2019). For MRPC and QQP, we report F1/Accuracy.

*then*

$$\epsilon^* \leq \epsilon_p$$

*where $\epsilon^*$ and $\epsilon_p$ denote the minimal generalization gaps one can achieve with or without augmented data, with at least $1 - \delta$ probability. If further assuming a better empirical risk with data augmentation, i.e., $\mathcal{R}_{emp}(f, g^*, \{\mathbf{x}_i\}_{i=1}^a \cup \{\mathbf{x}_i'\}_{i=1}^b) \leq \mathcal{R}_{emp}(f, g_p, \{\mathbf{x}_i\}_{i=1}^n)$, we have*

$$\mathcal{R}(f, g^*, p) \leq \mathcal{R}(f, g_p, p)$$

**Remark 4** *For Theorem 3 to hold, based on Baxter (2000), it is enough to ensure $\{\mathbf{x}_i, \mathbf{x}_i'\}$ and $\{\mathbf{x}_j, \mathbf{x}_j'\}$ to be independent for $i \neq j$. We achieve this by constructing $\mathbf{x}_i'$ with $\mathbf{x}_i$ and an extra random sample from the training data. Since all $(\mathbf{x}_i, \mathbf{x}_j)$ and the extra random samples are independent, the resulting concatenation will also be independent.*

## 4 EXPERIMENTS

We demonstrate the effectiveness of MixKD on a number of GLUE (Wang et al., 2019) dataset tasks: Stanford Sentiment Treebank (SST-2) (Socher et al., 2013), Microsoft Research Paraphrase Corpus (MRPC) (Dolan & Brockett, 2005), Quora Question Pairs (QQP)[3], Multi-Genre Natural Language Inference (MNLI) (Williams et al., 2018), Question Natural Language Inference (QNLI) (Rajpurkar et al., 2016), and Recognizing Textual Entailment (RTE) (Dagan et al., 2005; Haim et al., 2006; Giampiccolo et al., 2007; Bentivogli et al., 2009). Note that MNLI contains both an in-domain (MNLI-m) and cross-domain (MNLI-mm) evaluation set. These datasets span sentiment analysis, paraphrase similarity matching, and natural language inference types of tasks. We use the Hugging Face Transformers[4] implementation of BERT for our experiments.

### 4.1 GLUE DATASET EVALUATION

We first analyze the contributions of each component of our method, evaluating on the dev set of the GLUE datasets. For the teacher model, we fine-tune a separate 12 Transformer-layer `bert-base-uncased` ($\mathrm{BERT}_{12}$) for each task. We use the smaller $\mathrm{BERT}_3$ and $\mathrm{BERT}_6$ as the student model. We find that initializing the embeddings and Transformer layers of the student model from the first $k$ layers of the teacher model provides a significant boost to final performance. We use MSE as the knowledge distillation distance metric $d(\cdot, \cdot)$. We generate one mixup sample for each original sample in each minibatch (mixup ratio of 1), with $\lambda \sim \mathrm{Beta}(0.4, 0.4)$. We set hyperparameters weighting the components in the loss term in equation 8 as $\alpha_{\mathrm{SM}} = \alpha_{\mathrm{TMKD}} = 1$.

---

[3]`data.quora.com/First-Quora-Dataset-Release-Question-Pairs`
[4]`https://huggingface.co/transformers/`

| Model | SST-2 | MRPC | QQP | MNLI-m | MNLI-mm | QNLI | RTE |
|---|---|---|---|---|---|---|---|
| BERT$_{12}$ | 93.5 | 88.9/84.8 | 71.2/89.2 | 84.6 | 83.4 | 90.5 | 66.4 |
| BERT$_6$-FT | 90.7 | 85.9/80.2 | 69.2/88.2 | 80.4 | 79.7 | 86.7 | 63.6 |
| BERT$_6$-KD | 91.5 | 86.2/80.6 | 70.1/88.8 | 80.2 | 79.8 | 88.3 | 64.7 |
| BERT$_6$-PKD | 92.0 | 85.0/79.9 | **70.7**/88.9 | 81.5 | 81.0 | **89.0** | 65.5 |
| BERT$_6$-*MixKD* | **92.5** | **86.4/81.9** | 70.5/**89.1** | **82.2** | **81.2** | 88.2 | **68.3** |
| BERT$_3$-FT | 86.4 | 80.5/72.6 | 65.8/86.9 | 74.8 | 74.3 | 84.3 | 55.2 |
| BERT$_3$-KD | 86.9 | 79.5/71.1 | 67.3/87.6 | 75.4 | 74.8 | 84.0 | 56.2 |
| BERT$_3$-PKD | 87.5 | 80.7/72.5 | **68.1/87.8** | 76.7 | 76.3 | **84.7** | 58.2 |
| BERT$_3$-*MixKD* | **89.5** | **83.3/75.2** | 67.2/87.4 | **77.2** | **76.8** | 84.4 | **62.0** |

Table 3: GLUE test server results. We show results for the full variants of the 3- and 6-layer *MixKD* student models (SM+TMKD+BT). Knowledge distillation (KD) and Patient Knowledge Distillation (PKD) results are from Sun et al. (2019a).

As a baseline, we fine-tune the student model on the task dataset without any distillation or augmentation, which we denote as BERT$_k$-FT. We compare this against *MixKD*, with both knowledge distillation on the teacher's predictions ($\mathcal{L}_{\text{TMKD}}$) and mixup for the student ($\mathcal{L}_{\text{SM}}$), which we call BERT$_k$-SM+TMKD. We also evaluate an ablated version without the student mixup loss (BERT$_k$-TMKD) to highlight the knowledge distillation component specifically. We note that our method can also easily be combined with other forms of data augmentation. For example, backtranslation (translating an input sequence to the data space of another language and then translating back to the original language) tends to generate varied but semantically similar sequences; these sentences also tend to be of higher quality than masking or word-dropping approaches. We show that our method has an additive effect with other techniques by also testing our method with the dataset augmented with German backtranslation, using the `fairseq` (Ott et al., 2019) neural machine translation codebase to generate these additional samples. We also compare all of the aforementioned variants with backtranslation samples augmenting the data; we denote these variants with an additional +BT.

We report the model accuracy (and $F_1$ score, for MRPC and QQP) in Table 1. We also show the performance of the full-scale teacher model (BERT$_{12}$) and DistilBERT (Sanh et al., 2019), which performs basic knowledge distillation during BERT pre-training to a 6-layer model. For our method, we observe that a combination of data augmentation and knowledge

| Model | Inference Speed (samples/second) | # of Parameters |
|---|---|---|
| BERT$_{12}$ Teacher | 115 | 109,483,778 |
| BERT$_6$ Student | 252 | 66,956,546 |
| BERT$_3$ Student | 397 | 45,692,930 |

Table 2: Computation cost comparison of teacher and student models on SST-2 with batch size of 16 on a Nvidia TITAN X GPU.

distillation leads to significant gains in performance, with the best variant often being the combination of teacher mixup knowledge distillation, student mixup, and backtranslation. In the case of SST-2, for example, BERT$_6$-SM+TMKD+BT is able to capture $99.88\%$ of the performance of the teacher model, closing $91.27\%$ of the gap between the fine-tuned student model and the teacher, despite using far fewer parameters and having a much faster inference speed (Table 2).

After analyzing the contributions of the components of our model on the dev set, we find the SM+TMKD+BT variant to have the best performance overall and thus focus on this variant. We submit this version of *MixKD* to the GLUE test server, reporting its results in comparison with fine-tuning (FT), vanilla knowledge distillation (KD) (Hinton et al., 2015), and patient knowledge distillation (PKD) (Sun et al., 2019a) in Table 3. Once again, we observe that our model outperforms the baseline methods on most tasks.

## 4.2 LIMITED-DATA SETTINGS

One of the primary motivations for using data augmentation for knowledge distillation is to give the student more opportunities to query the teacher model. For datasets with a large enough number of samples relative to the task's complexity, the original dataset may provide enough chances to learn from the teacher, reducing the relative value of data augmentation.

As such, we also evaluate *MixKD* with a BERT$_3$ student on downsampled versions of QQP, MNLI (matched and mismatched), and QNLI in Figure 1. We randomly select $10\%$ and $1\%$ of the data from

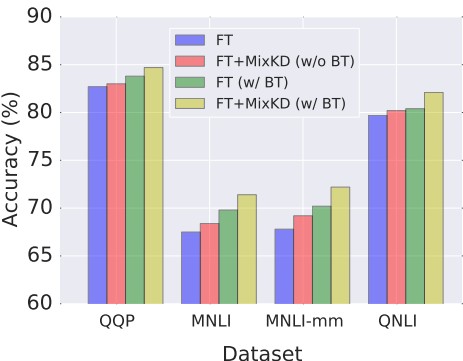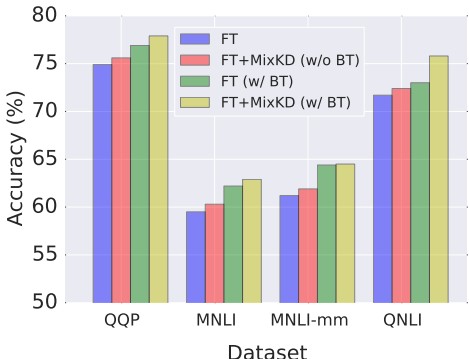

Figure 1: Results of limited data case, where both the teacher and student models are learned with only 10% (left) or 1% of the training data (right).

these datasets to train both the teacher and student models, using the same subset for all experiments for fair comparison. In this data limited setting, we observe substantial gains from *MixKD* over the fine-tuned model for QQP (+2.0%, +3.0%), MNLI-m (+3.9%, +3.4%), MNLI-mm (+4.4%, +3.3%), and QNLI (+2.4%, +4.1%) for 10% and 1% of the training data.

### 4.3 Embeddings Visualization

We perform a qualitative examination of the effect of the proposed *MixKD* by visualizing the latent space between positive and negative samples as encoded by the student model with t-SNE plots (Maaten & Hinton, 2008). In Figure 2, we show the shift of the transformer features at the [CLS] token position, with and without mixup data augmentation from the teacher. We randomly select a batch of 100 sentences from the SST-2 dataset, of which 50 are positive sentiment (blue square) and 50 are negative sentiment (red circle). The intermediate mixup neighbours are indicated by triangles with color determined by the closeness to the positive group or negative group. From Figure 2(a) to Figure 2(b), *MixKD* forces the linearly interpolated samples to be aligned with the manifold formed by the real training data and leads the student model to explore meaningful regions of the feature space effectively.

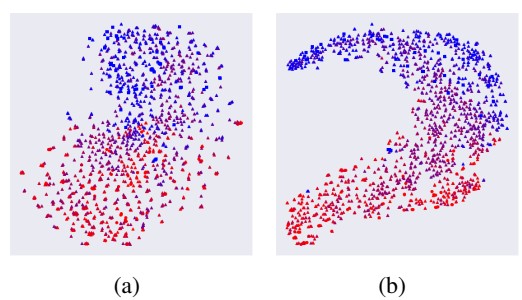

(a)                    (b)

Figure 2: Latent space of randomly sampled training data and their mixup neighbours encoded by student model (a) learned by standard fine-tuning (b) learned with *MixKD*.

### 4.4 Hyperparameter Sensitivity & Further Analysis

**Loss Hyperparameters** Our final objective in equation 8 has hyperparameters $\alpha_{\text{SM}}$ and $\alpha_{\text{TMKD}}$, which control the weight of the student model's cross-entropy loss for the mixup samples and the knowledge distillation loss with the teacher's predictions on the mixup samples, respectively. We demonstrate that the model is fairly stable over a wide range by sweeping both $\alpha_{\text{SM}}$ and $\alpha_{\text{TMKD}}$ over the range $\{0.1, 0.5, 1.0, 2.0, 10.0\}$. We do this for a $\text{BERT}_3$ student and $\text{BERT}_{12}$ teacher, with SST-2 as the task; we show the results of this sensitivity study, both with and without German backtranslation, in Figure 3. Given the overall consistency, we observe that our method is stable over a wide range of settings.

**Mixup Ratio** We also investigate the effect of the mixup ratio: the number of mixup samples generated for each sample in a minibatch. We run a smaller sweep of $\alpha_{\text{SM}}$ and $\alpha_{\text{TMKD}}$ over the range $\{0.5, 1.0, 2.0\}$ for mixup ratios of 2 and 3 for a $\text{BERT}_3$ student SST-2, with and without German backtranslation, in Figure 3. We conclude that the mixup ratio does not have a strong effect on overall performance. Given that higher mixup ratio requires more computation (due to more samples over which to compute the forward and backward pass), we find a mixup ratio of 1 to be enough.

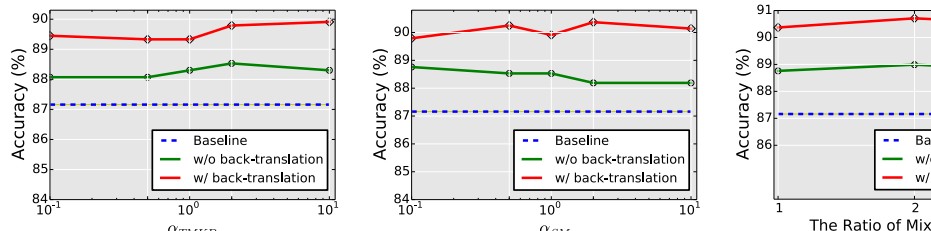

Figure 3: Hyperparameter sensitivity analysis regarding the *MixKD* approach, with different choices of $\alpha_{\text{TMKD}}, \alpha_{\text{SM}}$ and the ratio of mixup samples (*w.r.t.* the original training data).

**Comparing with TinyBERT's DA module** TinyBERT (Jiao et al., 2019) also utilizes data augmentation for knowledge distillation. Specifically, they adopt a conditional BERT contextual augmentation (Wu et al., 2019) strategy. To further verify the effectiveness of our approach, we use TinyBERT's released codebase[5] to generate augmented samples and make

| Methods | MNLI | SST-2 |
|---|---|---|
| $BERT_6$ | 81.3 | 90.9 |
| $BERT_6$ + TinyBERT DA module | 81.5 | 91.3 |
| $BERT_6$ + *MixKD* | **82.5** | **92.1** |

Table 4: We compare our approach with the data augmentation module proposed by TinyBert (Jiao et al., 2019).

a direct comparison with *MixKD*. As shown in Table 4, our approach exhibits much stronger results for distilling a 6-layer BERT model (on both MNLI and SST-2 datasets). Notably, TinyBERT's data augmentation module is much less efficient than mixup's simple operation, generating 20 times the original data as augmented samples, thus leading to massive computation overhead.

## 5 CONCLUSIONS

We introduce *MixKD*, a method that uses data augmentation to significantly increase the value of knowledge distillation for compressing large-scale language models. Intuitively, *MixKD* allows the student model additional queries to the teacher model, granting it more opportunities to absorb the latter's richer representations. We analyze *MixKD* from a theoretical standpoint, proving that our approach results in a smaller gap between generalization error and empirical error, as well as better generalization, under appropriate conditions. Our approach's success on a variety of GLUE tasks demonstrates its broad applicability, with a thorough set of experiments for validation. We also believe that the *MixKD* framework can further reduce the gap between student and teacher models with the incorporation of more recent mixup and knowledge distillation techniques (Lee et al., 2020; Wang et al., 2020; Mirzadeh et al., 2019), and we leave this to future work.

### ACKNOWLEDGMENTS

CC is partly supported by the Verizon Media FREP program.

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

## A  PROOFS

**Proof** [Proof of Theorem 1] First of all, $\{\mathbf{x}_i\}_{i=1}^a \cup \{\mathbf{x}_i'\}_{i=1}^b$ can be regarded as drawn from distribution $r(\mathbf{x}) = \dfrac{ap(\mathbf{x}) + bq(\mathbf{x})}{a + b}$.

Given $\mathcal{G}$ is finite, we have the following theorem

**Theorem 5** *(Mohri et al., 2018) Let $l$ be a bounded loss function, hypothesis set $\mathcal{G}$ is finite. Then for any $\delta > 0$, with probability at least $1 - \delta$, the following inequality holds for all $g \in \mathcal{G}$:*

$$\mathcal{R}(f, g, p) - \mathcal{R}_{emp}(f, g, \{\mathbf{x}_i\}_{i=1}^n) \leq M\sqrt{\frac{\log(|\mathcal{G}|/\delta)}{2n}}$$

Thus we have in our case:

$$\mathcal{R}(f, g_p, p) - \mathcal{R}_{emp}(f, g_p, \{\mathbf{x}_i\}_{i=1}^n) \leq \epsilon_p \leq M\sqrt{\frac{\log(|\mathcal{G}|/\delta)}{2n}}$$

and

$$
\begin{aligned}
&\mathcal{R}(f, g^*, p) - \mathcal{R}_{emp}(f, g^*, \{\mathbf{x}_i\}_{i=1}^a \cup \{\mathbf{x}_i'\}_{i=1}^b) \\
=&\mathcal{R}(f, g^*, r) - \mathcal{R}_{emp}(f, g^*, \{\mathbf{x}_i\}_{i=1}^a \cup \{\mathbf{x}_i'\}_{i=1}^b) + \int l(f(\mathbf{x}), g^*(\mathbf{x}))(p(\mathbf{x}) - r(\mathbf{x}))d\mathbf{x} \\
=&\mathcal{R}(f, g^*, r) - \mathcal{R}_{emp}(f, g^*, \{\mathbf{x}_i\}_{i=1}^a \cup \{\mathbf{x}_i'\}_{i=1}^b) + \frac{b}{a+b}\int l(f(\mathbf{x}), g^*(\mathbf{x}))(p(\mathbf{x}) - q(\mathbf{x}))d\mathbf{x} \\
\leq&\mathcal{R}(f, g^*, r) - \mathcal{R}_{emp}(f, g^*, \{\mathbf{x}_i\}_{i=1}^a \cup \{\mathbf{x}_i'\}_{i=1}^b) + \int l(f(\mathbf{x}), g^*(\mathbf{x}))(p(\mathbf{x}) - q(\mathbf{x}))d\mathbf{x} \\
\leq& M\sqrt{\frac{\log(|\mathcal{G}|/\delta)}{2(a+b)}} + \triangle
\end{aligned}
\tag{10}
$$

where $\triangle = \int l(f(\mathbf{x}), g^*(\mathbf{x}))(p(\mathbf{x}) - q(\mathbf{x}))d\mathbf{x}$. If

$$b \geq \frac{M^2 \log(|\mathcal{G}|/\delta)}{2(\epsilon_p - \triangle)^2} - a$$

then

$$
\begin{aligned}
2(a + b) &\geq \frac{M^2 \log(|\mathcal{G}|/\delta)}{(\epsilon_p - \triangle)^2} \\
(\epsilon_p - \triangle)^2 &\geq \frac{M^2 \log(|\mathcal{G}|/\delta)}{2(a+b)} \\
\epsilon_p &\geq M\sqrt{\frac{\log(|\mathcal{G}|/\delta)}{2(a+b)}} + \triangle
\end{aligned}
$$

Substitute into equation 10, we have

$$\mathcal{R}(f, g^*, p) - \mathcal{R}_{emp}(f, g^*, \{\mathbf{x}_i\}_{i=1}^a \cup \{\mathbf{x}_i'\}_{i=1}^b) \leq \epsilon_p$$

Recall the definition of $\epsilon^*$, which is the minimum value of all possible $\epsilon$ satisfying

$$\mathcal{R}(f, g^*, p) - \mathcal{R}_{emp}(f, g^*, \{\mathbf{x}_i\}_{i=1}^a \cup \{\mathbf{x}_i'\}_{i=1}^b) \leq \epsilon$$

we know that $\epsilon^* \leq \epsilon_p$. Let $c = 2(\epsilon_p - \triangle)^2$, we can conclude the theorem.

∎

**Proof** [Proof of Theorem 2] First of all, $\{\mathbf{x}_i\}_{i=1}^a \cup \{\mathbf{x}_i'\}_{i=1}^b$ can be regarded as drawn from distribution $r(\mathbf{x}) = \dfrac{ap(\mathbf{x}) + bq(\mathbf{x})}{a + b}$.

**Theorem 6** *(Mohri et al., 2018) Let $l$ be a non-negative loss function upper bounded by $M > 0$, and for any fixed $\mathbf{y}$, $l(\mathbf{y}, \mathbf{y}')$ is $L$-Lipschitz for some $L > 0$, then with probability at least $1 - \delta$,*

$$\mathcal{R}(f, g, p) - \mathcal{R}_{emp}(f, g, \{\mathbf{x}_i\}_{i=1}^n) \leq 2L\mathfrak{R}_p(\mathcal{G}) + M\sqrt{\frac{log(1/\delta)}{2n}}$$

Thus we have

$$\mathcal{R}(f, g, p) - \mathcal{R}_{emp}(f, g, \{\mathbf{x}_i\}_{i=1}^n) \leq \epsilon_p \leq 2L\mathfrak{R}_p(\mathcal{G}) + M\sqrt{\frac{log(1/\delta)}{2n}}$$

where $\mathfrak{R}_p(\mathcal{G})$ are Rademacher complexity over all samples of size $n$ samples from $p(\mathbf{x})$.

We also have

$$\mathcal{R}(f, g^*, p) - \mathcal{R}_{emp}(f, g^*, \{\mathbf{x}_i\}_{i=1}^a \cup \{\mathbf{x}'_i\}_{i=1}^b)$$

$$= \mathcal{R}(f, g^*, r) - \mathcal{R}_{emp}(f, g^*, \{\mathbf{x}_i\}_{i=1}^a \cup \{\mathbf{x}'_i\}_{i=1}^b) + \int l(f(\mathbf{x}), g^*(\mathbf{x}))(p(\mathbf{x}) - r(\mathbf{x}))d\mathbf{x}$$

$$= \mathcal{R}(f, g^*, r) - \mathcal{R}_{emp}(f, g^*, \{\mathbf{x}_i\}_{i=1}^a \cup \{\mathbf{x}'_i\}_{i=1}^b) + \frac{b}{a+b}\int l(f(\mathbf{x}), g^*(\mathbf{x}))(p(\mathbf{x}) - q(\mathbf{x}))d\mathbf{x}$$

$$\leq \mathcal{R}(f, g^*, r) - \mathcal{R}_{emp}(f, g^*, \{\mathbf{x}_i\}_{i=1}^a \cup \{\mathbf{x}'_i\}_{i=1}^b) + \int l(f(\mathbf{x}), g^*(\mathbf{x}))(p(\mathbf{x}) - q(\mathbf{x}))d\mathbf{x}$$

$$\leq 2L\mathfrak{R}_r(\mathcal{G}) + M\sqrt{\frac{log(1/\delta)}{2(a+b)}} + \triangle \tag{11}$$

where $\triangle = \int l(f(\mathbf{x}), g^*(\mathbf{x}))(p(\mathbf{x}) - q(\mathbf{x}))d\mathbf{x}$. $\mathfrak{R}_r(\mathcal{G})$ are Rademacher complexity over all samples of size $(a + b)$ samples from $r(\mathbf{x}) = \dfrac{ap(\mathbf{x}) + bq(\mathbf{x})}{a + b}$.

If

$$b \geq \frac{M^2 \log(1/\delta)}{2(\epsilon_p - \triangle - 2L\mathfrak{R}_r(\mathcal{G}))^2} - a$$

then:

$$2(a + b) \geq \frac{M^2 \log(1/\delta)}{(\epsilon_p - \triangle - 2L\mathfrak{R}_r(\mathcal{G}))^2}$$

$$\epsilon_p - \triangle - 2L\mathfrak{R}_r(\mathcal{G}) \geq M\sqrt{\frac{\log(1/\delta)}{2(a+b)}}$$

$$\epsilon_p \geq M\sqrt{\frac{\log(1/\delta)}{2(a+b)}} + \triangle + 2L\mathfrak{R}_r(\mathcal{G})$$

Substitute into equation 11, we have:

$$\mathcal{R}(f, g^*, p) - \mathcal{R}_{emp}(f, g^*, \{\mathbf{x}_i\}_{i=1}^a \cup \{\mathbf{x}'_i\}_{i=1}^b) \leq \epsilon_p$$

Recall the definition of $\epsilon^*$, which is the minimum value of all possible $\epsilon$ satisfying

$$\mathcal{R}(f, g^*, p) - \mathcal{R}_{emp}(f, g^*, \{\mathbf{x}_i\}_{i=1}^a \cup \{\mathbf{x}'_i\}_{i=1}^b) \leq \epsilon$$

we know that $\epsilon^* \leq \epsilon_p$. Let $c = 2(\epsilon_p - \triangle - 2L\mathfrak{R}_r(\mathcal{G}))^2$, we can conclude the theorem. ∎

**Proof** [Proof of Theorem 3] Similar to previous theorems, we write

$$\mathcal{R}(f, g^*, p) - \mathcal{R}_{emp}(f, g^*, \{\mathbf{x}_i\}_{i=1}^a \cup \{\mathbf{x}'_i\}_{i=1}^b)$$

$$= \mathcal{R}(f, g^*, \frac{ap + bq}{a+b}) - \mathcal{R}_{emp}(f, g^*, \{\mathbf{x}_i\}_{i=1}^a \cup \{\mathbf{x}'_i\}_{i=1}^b) + \int l(f(\mathbf{x}), g^*(\mathbf{x}))(p(\mathbf{x}) - \frac{ap(\mathbf{x}) + bq(\mathbf{x})}{a+b})d\mathbf{x}$$

$$= \mathcal{R}(f, g^*, \frac{ap + bq}{a+b}) - \mathcal{R}_{emp}(f, g^*, \{\mathbf{x}_i\}_{i=1}^a \cup \{\mathbf{x}'_i\}_{i=1}^b) + \frac{b}{a+b}\int l(f(\mathbf{x}), g^*(\mathbf{x}))(p(\mathbf{x}) - q(\mathbf{x}))d\mathbf{x}$$

$$\leq \mathcal{R}(f, g^*, \frac{ap + bq}{a+b}) - \mathcal{R}_{emp}(f, g^*, \{\mathbf{x}_i\}_{i=1}^a \cup \{\mathbf{x}'_i\}_{i=1}^b) + \triangle \tag{12}$$

where $\triangle = \int l(f(\mathbf{x}), g^*(\mathbf{x}))(p(\mathbf{x}) - q(\mathbf{x}))d\mathbf{x}$. For notation consistency, we write $\mathcal{R}(f, g^*, \frac{ap+bq}{a+b}) = \int l(f(\mathbf{x}) - g(\mathbf{x}))\frac{ap(\mathbf{x}) + bq(\mathbf{x})}{a+b}d\mathbf{x}$. However, $\{\mathbf{x}_i\}_{i=1}^a \cup \{\mathbf{x}_i'\}_{i=1}^b$ are not drawn from the same distribution (which is $r(\mathbf{x}) = \dfrac{ap(\mathbf{x}) + bq(\mathbf{x})}{a+b}$ in previous cases).

Let $\gamma = \lfloor \dfrac{a+b}{a} \rfloor$, we split $\{\mathbf{x}_i\}_{i=1}^a \cup \{\mathbf{x}_i'\}_{i=1}^b$ into $\gamma$ parts that don't overlap with each other. The first part is $\{\mathbf{x}_i\}_{i=1}^a$, all the other parts has at least $a$ elements from $\{\mathbf{x}_i'\}_{i=1}^b$.

Let

$$\lambda = \sqrt{\frac{64}{b}\log(4/\delta) + \frac{64}{a}\log C(\mathcal{G})}$$

where $C(\mathcal{G})$ is space capacity defined in Definition 4 in Baxter (2000), which depends on $\epsilon^*$ and $\mathcal{G}$. By Theorem 4 in Baxter (2000),

$$\left[\mathcal{R}(f, g^*, \frac{ap+bq}{a+b}) - \mathcal{R}_{emp}(f, g^*, \{\mathbf{x}_i\}_{i=1}^a \cup \{\mathbf{x}_i'\}_{i=1}^b)\right]^2 \leq \max\{\frac{64}{\gamma a}\log(\frac{4C(\mathcal{G}^\gamma)}{\delta}), \frac{16}{a}\}$$

By Theorem 5 in Baxter (2000),

$$\frac{64}{\gamma a}\log(\frac{4C(\mathcal{G}^\gamma)}{\delta}) = \frac{64}{\gamma a}(\log(\frac{4}{\delta}) + \log(C(\mathcal{G}^\gamma))) \leq \frac{64}{\gamma a}(\log(\frac{4}{\delta}) + \gamma\log(C(\mathcal{G}))) \leq \lambda^2$$

The last inequality comes from $b \leq \gamma a$, which is because of $\gamma = \lfloor \dfrac{a+b}{a} \rfloor$. Then we have

$$\left[\mathcal{R}(f, g^*, \frac{ap+bq}{a+b}) - \mathcal{R}_{emp}(f, g^*, \{\mathbf{x}_i\}_{i=1}^a \cup \{\mathbf{x}_i'\}_{i=1}^b)\right]^2 \leq \max\{\frac{64}{\gamma a}\log(\frac{4C(\mathcal{G}^\gamma)}{\delta}), \frac{16}{a}\} \leq \max\{\lambda^2, \frac{16}{a}\}$$

If

$$b \geq \frac{64\log(4/\delta)}{(\epsilon_p - \triangle)^2 - 64\log C(\mathcal{G})/a}$$

Then

$$\lambda^2 \leq \frac{64}{a}\log C(\mathcal{G}) + 64\log(\frac{4}{\delta})\frac{(\epsilon_p - \triangle)^2 - 64\log C(\mathcal{G})/a}{64\log(4/\delta)}$$

$$\lambda^2 \leq (\epsilon_p - \triangle)^2 \tag{13}$$

If

$$\frac{16}{(\epsilon_p - \triangle)^2} \leq a$$

then

$$\frac{16}{a} \leq (\epsilon_p - \triangle)^2 \tag{14}$$

Combine equation 13 and equation 14, we have

$$\mathcal{R}(f, g^*, \frac{ap+bq}{a+b}) - \mathcal{R}_{emp}(f, g^*, \{\mathbf{x}_i\}_{i=1}^a \cup \{\mathbf{x}_i'\}_{i=1}^b) \leq \epsilon_p - \triangle$$

Substitute into equation 12, we have:

$$\mathcal{R}(f, g^*, p) - \mathcal{R}_{emp}(f, g^*, \{\mathbf{x}_i\}_{i=1}^a \cup \{\mathbf{x}_i'\}_{i=1}^b) \leq \epsilon_p$$

Recall the definition of $\epsilon^*$, which is the minimum value of all possible $\epsilon$ satisfying

$$\mathcal{R}(f, g^*, p) - \mathcal{R}_{emp}(f, g^*, \{\mathbf{x}_i\}_{i=1}^a \cup \{\mathbf{x}_i'\}_{i=1}^b) \leq \epsilon$$

we know that $\epsilon^* \leq \epsilon_p$.

■

# B    VARIANCE ANALYSIS

For the purpose of getting a sense of variance, we run experiments with additional random seeds on MRPC and RTE, which are relatively smaller datasets, and MNLI and QNLI, which are relatively larger datasets. Mean and standard deviation on the dev set of these GLUE datasets are reported in Table 5. We observe the variance of the same model's performance to be small, especially on the relatively larger datasets.

| Model | MRPC | MNLI-m | QNLI | RTE |
|---|---|---|---|---|
| $BERT_6$-TMKD+BT | **89.79**$\pm$0.27/**85.04**$\pm$0.48 | 82.05$\pm$0.11 | 88.42$\pm$0.06 | **69.37**$\pm$0.50 |
| $BERT_6$-SM+TMKD+BT | 89.64$\pm$0.38/84.43$\pm$0.36 | **82.41**$\pm$0.12 | **88.76**$\pm$0.15 | 68.02$\pm$0.11 |
| $BERT_3$-TMKD+BT | **84.79**$\pm$0.33/**75.82**$\pm$0.48 | 77.16$\pm$0.03 | 84.60$\pm$0.07 | **62.47**$\pm$0.36 |
| $BERT_3$-SM+TMKD+BT | 84.53$\pm$0.39/75.85$\pm$0.60 | **77.42**$\pm$0.11 | **84.88**$\pm$0.06 | 60.83$\pm$0.18 |

Table 5: Mean and variance reported for $BERT_6$-TMKD+BT,$BERT_6$-SM+TMKD+BT,$BERT_3$-TMKD+BT and $BERT_3$-SM+TMKD+BT.

