# OpenReview forum: "MixKD: Towards Efficient Distillation of Large-scale Language Models"
_ICLR.cc/2021/Conference — ICLR 2021 Poster_

### Official Review · AnonReviewer4 · 2020-10-25
**An interesting paper but not good enough**

**Rating:** 5
**Confidence:** 4

**Review:**

This paper applies mixup (Zhang et al., 2018) to augment training data to improve knowledge distillation in NLP tasks . Mixup was originally proposed to augment data for continuous data. To apply mixup to textual data, this paper applies mixup to the word/token embeddings instead of the tokens themselves. Some theoretical analysis has been done, and the experimental results show improved metrics over baseline methods such as DistillBERT.


*Strong points*
- This paper is well-written and very easy to follow.
- Theoretical results have been provided and the experiment has been carefully done.

*Weak points*
- The proposed method is a direct application of mixup (Zhang et al., 2018). The overall idea is quite straightforward.
- The empirical results of the proposed method are not impressive (worse than the metrics reported in TinyBERT (Jiao et al., 2019)?).
- It is true that for benchmarks like GLUE, the datasets are quite small so that data augmentation is important for knowledge distillation. This is however not true for most real-world applications where we often have almost unlimited unlabelled data (e.g., from logs, extract from webs etc). In other words, text data augmentation for knowledge distillation is not necessarily an important problem for real-world applications.

*Questions & Other comments*
- In Table 3, the metrics for TinyBERT is different from what has been reported in TinyBERT (Jiao et al., 2019). Can you add more explanation here?
- “Notably, TinyBERT’s data augmentation module is much less efficient than mixup’s simple operation” It is not an issue to have some computation overhead in data augmentation, as it is a one-off operation.

---

> ### Author Response · Authors · 2020-11-21
> **Reviewer 4 Response**
>
> We would like to thank reviewer 4 for the thoughtful comments. Below we address the concerns mentioned in the review:
>
> **Importance of data augmentation for text data**
> We push back on the notion that "text data augmentation for knowledge distillation is not necessarily an important problem for real-world applications." Please note that our method focuses on the supervised problem of task-specific fine-tuning, not the pre-training stage. As such, we focus on labeled data, not unlabeled data. While we do agree that there is indeed a vast wealth of text data available in certain contexts, this is certainly not the case in general. There are many applications where data may be quite limited: a few examples include doctor notes for rare diseases, small business analytics (e.g emails, job descriptions), anomaly detection, low-resource languages, and few-shot learning. As such, we believe there is still great value in text data augmentation for real-world applications.
>
> **TinyBERT**
> While TinyBERT is certainly a relevant work, we caution against direct comparisons between the two. TinyBERT performs distillation in two-stages: pre-training and task-specific finetuning. MixKD only does the latter, which makes it significantly easier to use in practice, as large language model pre-training can be computationally expensive (16 TPUs for 4 days for BERT base). TinyBERT also uses a multilayer distillation strategy, while we again use a simpler approach of just distilling the logits. That said, these ideas from TinyBERT can be readily combined with MixKD to push our numbers higher. However, the goal of our paper was not necessarily to optimize the absolute state-of-the-art, but rather to prove that our simple approach leads to significant improvements in compressed model performance.
>
> **Metrics**
> In the TinyBERT paper, we did not find the numbers for the setting where only the data augmentation module is applied to the baseline. Therefore, to make a fair comparison between MixKD and the data augmentation module proposed in TinyBERT, we use the released code of TinyBERT to generate the augmented samples and reproduce the corresponding experiments.
>
> **Efficiency of the data augmentation module**
> We agree that the computation time for producing augmented samples does not matter that much, since we can do it offline. However, in TinyBERT, the ratio of augmented samples to original data points is set to 20, on average. This will greatly enlarge the training set size, thus taking much longer time to train relative to MixKD (where only one augmented sentence is generated from each original input). While longer training time due to data augmentation doesn't impact inference time performance, it does have significant consequences with respect to development time and iterability.

---

### Official Review · AnonReviewer3 · 2020-10-27

**Rating:** 7
**Confidence:** 3

**Review:**

The paper proposes combining the MixUp data augmentation method with teacher-student distillation to improve the fine-tuned performance of BERT on benchmark NLP tasks (GLUE). The problem is important, well-motivated and of interest to a broad base of NLP researchers and practitioners. The paper is clear, and generally well-written, although the idea itself is not surprisingly novel (somewhat of a low-hanging fruit), from the experimental results, the method improves upon baselines, and so the real-world impact could be high, especially given its simple implementation.

Pros:
* Well-written, solid experiments / ablations on a canonical benchmark (bonus points for reporting GLUE test set performance), some theoretical contributions.
* Reports performance in labeled-data-limited setting (by subsampling GLUE dataset)

Cons:
* In the theory, a bounded loss function is assumed, but in practice the unbounded cross-entropy is used (right?)
* Optimal hyper-parameters $\alpha_\text{SM}$ and $\alpha_\text{TMKD}$ for "MixKD" on the GLUE dev test are not reported in the main text
* "MixKD" includes a back-translation (BT) loss term as well as a student loss on mixup samples (SM). While the impact of each term is studied on the GLUE dev set, I'm interested in seeing its performance on GLUE test.
* In the theorems, $\delta$ should be re-introduced.
* The embedding visualization section needs improvement. What is being visualized here? 2-dimensional logits / probabilities from the model or input embeddings projected down to 2D? Furthermore, it is hard to distinguish circles from triangles without zooming in 3-5x.

---

> ### Author Response · Authors · 2020-11-21
> **Reviewer 3 Response**
>
> Thank you for the positive review! To answer your concerns:
>
> C1. In practice, cross-entropy loss is often calculated based on the softmax output, with the ground truth as a one-hot vector. This means the loss will not exactly reach 0, thus the logarithm term in cross-entropy will always be bounded, leading to a bounded loss. In addition, we may also add a small value, 1e-7 for example, to all the outputs of softmax for stability. This also makes the cross-entropy loss bounded, without influencing the prediction too much.
>
> C2. Thanks for the catch; we'll add it to our draft. We used $\alpha_{SM}=1$ and $\alpha_{TMKD}=1$. As shown in our hyperparameter sensitivity analysis in Figure 3, we didn't find the exact value of these hyperparameters to matter too much.
>
> C3. The reason we did our thorough ablation analysis on the dev set is because the labels are not publicly released for the test set. Calculating GLUE test performance requires submitting to the test server, which imposes a limit to how many submissions can be done.
>
> C4. Thanks for the suggestion; we will change it accordingly.
>
> C5. It's a t-SNE plot of the transformer features at the [CLS] token position. Sorry that the figure was too small. We'll expand it in our next draft.

---

### Official Review · AnonReviewer2 · 2020-10-27
**Intuitive approach to improve the benefits gained from KD**

**Rating:** 6
**Confidence:** 3

**Review:**

Nice paper. The main idea in this paper is to use a specific kind of data augmentation, Mixup (Manifold Mixup), in order to improve the effectiveness of the KD process and obtain better performing student models, especially in cases where not enough data is available on the target dataset and task.

While the idea is interesting in itself, I think what is proposed in this paper, is very relevant to this paper (noisy student): https://arxiv.org/abs/1911.04252

So, methodologically it's not super novel, but it should also be taken into account that many of the developments in this area have happened very recently. Moreover, the paper still has an added value since it is applying these techniques for a different input modality and in a slightly different setup.

Summary of the experiments:
-12 layer BERT is fine tuned as the teacher (different fine-tuned teacher for each task).
- Two different architectures are used for student models (6 layer BERT and 3 layer BERT).
- Student models are initialised by copying the lower layers of the teachers.
- The teacher is used to provide pseudo labels for the student model on the target dataset, while the target dataset is augmented with the manifold mixup approach on the embedding layer (Proposed Approach).
- The proposed approach is compared with plain KD and no KD (just fine-tuning on target).

My Questions:
- During KD, do you feed original examples from target to the teacher then apply mixup, or do you first apply mixup then feed the generated examples to the teacher and get the pseudo labels from the teacher?
-Most importantly, It is not clear to me from the experiments whether there is still an advantage of doing this, if the teacher is trained well enough e.g., with the same type of augmentation (intuitively, if the teacher is trained well enough and KD process is well tuned, all the information that the model can gain by doing additional data augmentations should already be transferrable through the soft targets from the teacher.)

Some positive point about the paper:
- Ablation experiments are conducted to separate the gain from simple KD with KD+Mixup.

Some of my concerns:
- The improvements in the accuracies reported seem marginal and I think there are indication of the significancy of the results in the paper (e.g., mean and variance over several trials?).

Some minor points:
- Figure 2: Mixup and Non Mixup examples (circles and triangles) are not clearly differentiable (just hard to see).
-I think the type of data augmentation applied is a special case of Manifold Mixup, so it would be nice to cite this paper as well:  https://arxiv.org/abs/1806.05236

---

> ### Author Response · Authors · 2020-11-21
> **Reviewer 2 Response**
>
> We're glad you enjoyed our paper! To answer your questions:
>
> Q1: The latter: we perform mixup of the samples at the word embedding level and then feed them to the teacher for pseudolabels. Please refer to equations 2 and 4 in our paper.
>
> Q2: The motivation stems from the notion that if there is a limited dataset, then the teacher can only generate a limited set of pseudolabels for the student to learn from. With only a limited set, even if the teacher learns a richer structure, the student may not have enough opportunities to learn it from the teacher. Performing data augmentations allows the teacher to provide more opportunities for the student to distill the teacher networks' knowledge.
>
> Mean and variance: In our initial experiments, we found that especially on relatively larger datasets such as MNLI, QNLI and SST-2, the variance of the same model's performance is pretty small. Thus, the improvements from MixKD are consistent and demonstrate the advantages of our proposed approach. That said, it is a good idea to add variance analysis to our experiments.
>
> We'll expand Figure 2 to make it clearer, and we're happy to add a reference to Manifold Mixup.

---

> > ### Comment · AnonReviewer2 · 2020-11-21
> > **Thank you for your response.**
> >
> > Thanks you for your response and for clarifying these points.
> >
> > About Q2., to clarify my question, I think KD in it self is (potentially) such a powerful technique that the knowledge transferred from the teacher to the student can go beyond the examples used in the distillation process. And even with a very small number of examples it should be possible to transfer the knowledge of the teacher to the student. So, my question is more in the direction of if using this type of data augmentation for KD adds something fundamentally different to the KD process or it is mainly easing the process of transferring the knowledge. In the later case, for example, one should be able to achieve the same level of performance through simple KD,  e.g., by perfectly tuning the parameters of the KD process (e.g., softmax temperature).
> >
> > Also, another question about the variance analysis parts, do you see less variance when you apply mixup+KD compared to only KD?

---

> > > ### Author Response · Authors · 2020-11-24
> > > **Clarifications**
> > >
> > > From an intuition perspective: taking your suggestion of a “very small number of examples” to the extreme, if there were just 2 samples, the student would get very limited information on the features learned by the teacher using just KD. Given neural networks’ well documented ability to memorize samples, it’s possible that the student may overfit. With data augmentation, however, the student also gets the teacher’s predictions for the synthetic samples to learn from as well. For example, mixup can produce samples at the decision boundary (e.g. if $\lambda=0.5$), which can yield insights into the teacher’s decision process. Moreover, the intuition here is similar to how data augmentation helps NLP tasks in general, where the augmented examples can endow the learned model with stronger generalizability (especially when the number of training examples is relatively small).
> > >
> > > From an empirical perspective: note that we do compare against vanilla KD in Table 2, as well as PKD, which is a method that uses knowledge distillation across multiple layers. In general, we see better results on GLUE with our approach. We haven’t had the chance yet to do a thorough comparison of the variance analysis between MixKD and KD, but it’s something we’ll look into.

---

### Official Review · AnonReviewer1 · 2020-10-29
**Extensive experiments demonstrating effectiveness of distillation framework for large-scale NLP models**

**Rating:** 6
**Confidence:** 4

**Review:**

The paper describes a distillation framework that leans heavily on Mixup, an effective data augmentation technique that is typically applied to images, to achieve impressive results on a range of GLUE benchmarks (). By using SM+TMKD+BT, the student model (BERT_6) is able to capture 99.88% of the performance of the teacher model (BERT_12).

The authors provide good motivation for the problem of distilling large-scale language models and conduct extensive experiments across multiple GLUE benchmarks, including ablation and hyperparameter sensitivity studies, and the results are quite convincing. Figure 3 nicely demonstrates that the framework is not overly sensitive to any of its hyperparameters.

Questions:
- Given that the paper motivates distillation as a means of reducing the computational cost of state-of-the-art language models (less power, less memory, lower latency), could you also describe the number of parameters that the student models use and their latency at inference time (vs. that of the teacher models)?

While the paper is generally well-written, there are a number of small issues / inconsistencies:
- Nit: Mixup is inconsistently capitalized / hyphenated.
- Since Mixup is central to this distillation framework, it would be valuable to briefly describe the method in the body of the paper, before describing how it was adapted to the domain of language.
- “that a a multi-task BERT model…”
- “around an training example”
- Nit: “vicinal” is an extremely rare word, so it might be worth using a more accessible word such as “neighboring”
- Nit: “tabular data were also shown, to demonstrate generality” (remove comma)
- Nit: “the extra word embeddings are mixuped with zero paddings” (maybe “mixed up” is better? :))
- In Table 1, for ease of reading, it would be good to describe in the caption the meaning of the first and second ‘/’ delimited number in each cell.
- For Figure 2, please describe the dimensionality reduction method used to visualize the latent space of training data and augmented examples. The pattern looks compelling, but I don’t see you describe anywhere how the data was projected.

---

> ### Author Response · Authors · 2020-11-21
> **Reviewer 1 Response**
>
> We appreciate the positive review! Thank you for the miscellaneous suggestions; we'll incorporate them into our next draft.
>
> Computational costs: Good point, thanks for the suggestion. While the inference speed will depend on a few factors (hardware, dataset, batch size, etc.), for SST-2 with a batch size of 16:
> - $\text{BERT}_{12}$ Teacher Inference Speed: ~115 samples/second
> - $\text{BERT}_{6}$ Student Inference Speed: ~252 samples/second
> - $\text{BERT}_{3}$ Student Inference Speed: ~397 samples/second
> - $\text{BERT}_{12}$ Teacher Parameters: 109,483,778
> - $\text{BERT}_{6}$ Student Parameters: 66,956,546
> - $\text{BERT}_{3}$ Student Parameters: 45,692,930

---

### Public Comment · ~Vikas_Verma1 · 2020-11-11
**Interesting work! Idea is similar to previous work in Semi-supervised learning**

Dear Authors,

The idea is similar to previous work on semi-supervised learning, so you might consider discussing the similarities and differences wrt the following papers:

Interpolation consistency training: https://www.ijcai.org/Proceedings/2019/0504.pdf
MixMatch: https://arxiv.org/abs/1905.02249
GraphMix https://arxiv.org/abs/1909.11715

---

> ### Author Response · Authors · 2020-11-14
> **Thanks for the comment!**
>
> Hi Vikas, thanks for your interest in our work! We'd like to point out that our work doesn't explicitly consider semi-supervised learning, as it wasn't our main focus, though we certainly could. That said, we'll happily discuss the related work that you suggested in a future version.

---

> > ### Comment · AnonReviewer2 · 2020-11-21
> > **+1 to the related works in semi-supervised learning**
> >
> > I agree that these papers are very relevant to your proposed approach. That's correct that you are not phrasing your method as a semi-supervised learning approach, but I think your proposal, from both technical and motivational point of view is tightly related to these works in semi-supervised learning, i.e., Enabling the model to learn from fewer examples. However I think, we should note the main difference that is in your work the focus is on a KD setup where teacher and student models can be completely independent and the student model is (presumably) not exposed to any labeled (ground truth) example.
> >
> > To add to this list:
> > https://arxiv.org/abs/2001.07685

---

### Public Comment · ~Mingyang_Yi1 · 2020-11-14
**A very interesting paper, but potentially unreasonable assumption?**


This paper is very impressive. However, I have one question about the theoretical part. To obtain Theorem 1, one must first obtain equation (8). Equation (8) is based on the classical generalization error bound within finite hypothesis space, under the condition of i.i.d training samples. Specifically, in the paper, $\{x_{i}\}, i=1,\cdots,a$ and $x_{i}^{\prime}, i=1,\cdots, b$ can be viewed as $a + b$ samples drawn from distribution $r(x)$, and equation (8) is obtained based on the assumption of independence between $x_{i}$ and $x_{i}^{\prime}$. However, $x_{i}^{\prime}$ is an augmented sample generated from some raw data $x_{i}$ (by mix-up) and it is not reasonable to assume the independence between them. This problem exists in all three theorems in the theoretical part in Section 3.4.

---

> ### Author Response · Authors · 2020-11-14
> **thanks for your comment**
>
> Dear Mingyang,
>
> Thanks for your comments. Yes, we agree x_i and x'_i might not be iid. That is why we considered 3 cases. Case 3 (corresponding to Theorem 3) explicitly consider the non-iid case. Theorem 1 and 2 consider the case of existing a distribution such that x_i and x'_i are iid sample from it. The reason we consider this is that even though x_i and x'_i might be dependent, there may still exists a distribution whose iid samples coincide with x_i and x'_i.

---

> > ### Public Comment · ~Mingyang_Yi1 · 2020-11-15
> > **Thanks for replying, but still unreasonable?**
> >
> > Thanks for replying, but I'm still confused with Theorem 3. As claimed in the paper, the proof of Theorem 3 is based on Theorem 4 in Baxter (2000). The main idea of it is dividing the augmented data into a number of subset. Data in each subset is drawn from some distribution $P_{i}$. However, in Theorem 4 of Baxter  (2000), one should assume $z_{i,m}$ are i.i.d drawn from $P_{i}$. It means one should assume $x_{i}^{\prime}$ is independent with each other which is also an unreasonable assumption.

---

> > > ### Author Response · Authors · 2020-11-16
> > > **about independency**
> > >
> > > Dear Mingyang,
> > >
> > > Thanks for the comment. We agree augmented data samples should be independent from each other, which is used in proving Theorem 4 in Baxter 2000 (see Theorem 18 in Baxter 2000).
> > >
> > > In our method, we can actually make this requirement satisfied by manipulating the construction of augmented data. For example, suppose we need m augmented data for training in one iteration, the augmented data can be constructed with 2m true data samples as: x'_i = \lambda x_{2i-1} + (1 - \lambda) x_{2i}. This ensures the independence between x'_1,...,x'_m. Across different iterations, the independence is also easy to verify: for any two x'_i, x'_j, p(x'_i)p(x'_j)=p(x'_i, x'_j) always holds. Thus the independence between augmented data samples can be easily satisfied during the whole training process.
> > >
> > > We acknowledge our result does not apply to all data augmentations but will not cause practical issues. We will make this clear in our revision.

---

> > > > ### Public Comment · ~Mingyang_Yi1 · 2020-11-16
> > > > **About independency**
> > > >
> > > > Thanks for the quick replying. The construction of the independent subset is very impressive. However, one can not go through the Theorem 3 even with this construction. As you can see, the proof is based on Theorem 18 in Baxter 2000, which relies on Lemma 22 in Baxter 2000. In the proof of Lemma 22, they use Hoeddfing's inequality (see the first equation of page 184). Hence, it requires $z_{i,j}$ to be independent with each other. To apply it in this paper, one should assume that all the augmented data $x_{i}^{\prime}$ are independent. But your construction can not meet this requirement.

---

> > > > > ### Author Response · Authors · 2020-11-17
> > > > > **independency**
> > > > >
> > > > > Please note our way of constructing the augmented data described previously has already guaranteed that the augmented data are independent with each other.

---

> > > > > > ### Public Comment · ~Mingyang_Yi1 · 2020-11-17
> > > > > > **Not enough true sample**
> > > > > >
> > > > > > Thanks for the quick replying. Your construction requires $n\geq 2m$, then the number of augmented data does not always meet your requirement i.e. the equation above equation (11). Besides, to use the Theorem 18 in Baxter 2000, all the training samples ($x_{i}$ and $x_{i}^{\prime}$) should be independent of each other.

---

> > > > > > > ### Author Response · Authors · 2020-11-20
> > > > > > > **not a concern**
> > > > > > >
> > > > > > > Thanks for the comment. We would like to point out that for the theorem to hold, it is enough to ensure [x_i, x'_i] and [x_j, x'_j] to be independent, where *'* means augmented data. From our construction, this is easily satisfied. Regarding the number of augmented data, yes, it needs to be sufficiently large. That is why we need the condition of b in Theorem 3. In the practical case of large data, this will always be satisfied. We will make this point more clear.

---

### Decision · Program_Chairs · 2021-01-07
**Final Decision**

**Decision:**

Accept (Poster)

**Comment:**

This work explores the distillation of language models using MixUp for data augmentation. Distillation with MixUp seems to be novel in the narrow context of distilling language models, although it has been used before in different contexts as the reviewers point out. The results of the experimental validation are encouraging, and the application is valuable and of wide interest to the ICLR audience. I therefore recommend accepting this paper for a poster presentation.